# Enhanced Electrochemical Performance of Carbon Nanotube with Nitrogen and Iron Using Liquid Phase Plasma Process for Supercapacitor Applications

**DOI:** 10.3390/ijms19123830

**Published:** 2018-11-30

**Authors:** Heon Lee, Byung-Joo Kim, Sun-Jae Kim, Young-Kwon Park, Sang-Chul Jung

**Affiliations:** 1Department of Environmental Engineering, Sunchon National University, Suncheon 57922, Korea; honylee@hanmail.net; 2R&D Division, Korea Institute of Carbon Convergence Technology, Jeonju 54853, Korea; kimbj2015@gmail.com; 3Faculty of Nanotechnology and Advanced Materials Engineering, Sejong University, Seoul 05006, Korea; sjkim1@sejong.ac.kr; 4School of Environmental Engineering, University of Seoul, Seoul 02504, Korea; catalica@uos.ac.kr

**Keywords:** supercapacitor, iron oxide nanoparticle, nitrogen doped, carbon nanotube, liquid phase plasma

## Abstract

Nitrogen-doped carbon nanotubes (NCNTs) and iron oxide particles precipitated on nitrogen-doped carbon nanotubes (IONCNTs) were fabricated by a liquid phase plasma (LPP) process for applications to anode materials in supercapacitors. The nitrogen element and amorphous iron oxide nanoparticles were evenly disseminated on the pristine multiwall carbon nanotubes (MWCNTs). The electrochemical performance of the NCNTs and IONCNTs were investigated and compared with those of pristine MWCNTs. The IONCNTs exhibited superior electrochemical performance to pristine MWCNTs and NCNTs. The specific capacitance of the as-fabricated composites increased as the content of nitrogen and iron oxide particles increased. In addition, the charge transfer resistance of the composites was reduced with introducing nitrogen and iron oxide.

## 1. Introduction

The electrochemical capacitor (also called supercapacitor) has attracted considerable interest in recent years because it shows rapid recharge, long-term cycling performance, and high-power density [1,2,3]. Carbon-based materials (e.g., activated carbon, carbon nanotubes (CNTs), graphene, etc.) have excellent physicochemical and electrical performance and have attracted attention as electrode materials in supercapacitors. In particular, CNTs have nanostructures and excellent chemical, mechanical, thermal properties, and electrical properties [4,5,6]. On the other hand, they have low specific capacitance compared to other carbon-based materials like porous carbon materials. It is generally known that the specific capacitance of CNTs is about 100 F/g, which is approximately half that of porous carbon, such as activated carbon (200 F/g) [7,8]. This is because CNTs have lower specific areas than activated carbon due to a deficiency of micropores. Therefore, it is essential to improve the specific capacitance of CNTs before they can be applied as electrode materials in supercapacitors. Several attempts have been made to dope CNTs with heteroatoms or precipitate transition metal oxides on the surface of CNTs.

Doping the CNT surface with heteroatoms, such as nitrogen and boron, is an effective method to improve the specific capacitance of CNTs [9,10,11]. Heteroatoms on the surface of CNTs can cause electron modulation and increase their conductivity, which improves their specific capacitance [12,13]. In particular, the specific capacitance of nitrogen-doped, carbon-based materials were approximately two times higher than that of untreated materials [14]. In addition, the low concentration of heteroatoms, such as nitrogen, are effective in improving the specific capacitance [15].

Transition metal oxides are also attractive materials for improving the specific capacitance of CNTs. The transition metal oxide on the surface of carbon materials can result in fast faradaic redox reactions with the electrolyte, and potentially store more energy than traditional carbon materials [16,17]. Ruthenium oxide, which is a transition metal oxide, exhibits outstanding pseudocapacitive behavior and charge storage capacity [18,19,20]. On the other hand, it has several disadvantages, such as high cost, toxicity, and scarcity [21]. Among other transition metal oxides, iron oxide has attracted considerable interest as an alternative transition metal oxide due to earth abundance, low cost, and non-toxicity [22,23].

Recently, liquid phase plasma (LPP) has emerged as a useful method for fabricating metal nanoparticles from metal precursors in solution [24,25]. The LPP provides activated chemical species (excited hydrogen and oxygen) and a high energy reaction field so that the metal ion in the solution can be converted to metal particles [26,27,28]. Furthermore, metal nanoparticles can precipitate on the supporting carbonaceous material via an LPP reaction without any process or reducing agent [29,30].

This paper introduces a new strategy, the LPP process, which can fabricate nitrogen-doped carbon nanotubes (NCNT) and iron oxide particles precipitated on nitrogen-doped carbon nanotubes (IONCNTs) simply, to improve the electrochemical performance of multiwall carbon nanotubes (MWCNTs) used as electrode materials in supercapacitors. The effects of the composition, morphology, and chemical state of the as-fabricated composites were examined by instrument analysis. Moreover, the electrochemical performances of the as-fabricated composites were assessed by cyclic voltammetry and galvanostatic methods.

## 2. Results and Discussion

### 2.1. Characteristics of the As-Prepared Composites

Figure 1 presents field emission scanning electron microscope (FE-SEM) photograph and element mapped results of IONCNT-10 that had been prepared by the LPP process using 10 mM of iron chloride as a precursor of an iron particle with NCNT. The green, yellow, and blue dots indicated the nitrogen, iron, and oxygen, respectively, generated by the LPP process on the surface of MWCNT. The results suggest that the nitrogen and iron ions in the reactant solutions were converted to the nitrogen dopant and iron particles by the activated chemical species from the LPP reaction. In addition, it was shown that the nitrogen and iron elements were well distributed over the surface of the MWCNTs.

Table 1 shows the elemental compositions of the MWCNTs and as-fabricated CNTs measured by energy dispersive spectrometer (EDS) attached to field emission scanning electron microscope (FE-SEM). IONCT-5 and IONCT-10 are composites prepared by using iron chloride at 5 mM and 10 mM, respectively, to deposit iron particles on the surface of nitrogen-doped carbon nanotubes. The pristine MWCNTs consisted of 94.94% carbon and 5.06% oxygen in weight %. The NCNTs prepared by the LPP reaction contained 1.49 wt.% nitrogen. This suggests that the ammonium ions in the aqueous solution generated from the precursor were converted to nitrogen by the LPP reaction. The IONCNTs consisted of carbon, nitrogen, iron, and oxygen. The amount of iron in the as-fabricated CNTs was strongly affected by the concentration of the iron precursor in the aqueous reactant solution. The Fe^2+^ ions, which dissociated from the iron precursor, was reduced by the electrons generated by the LPP reaction and was converted to iron particles that precipitated on the surface of the NCNTs. In addition, the quantity of oxygen in the NCNTs was 5.42 wt.%, which was higher than that of the pristine MWCNTs (5.06 wt.%) but lower than that of IONCNT-5 and 10 (6.03 and 6.47 wt.%). The MWCNT surface and iron metal particles produced were partially oxidized during the LPP reaction. Previous studies [22,24,25] showed that the metal nanoparticles generated by the LPP reaction from the metal precursor solution existed in the form of metal oxide. Various oxidative species, such as hydroxyl radicals and excited oxygen, were generated along with excited hydrogen in the plasma field during the LPP reaction, which partially led to the oxidation reaction of the surrounding substances.

The morphology of the iron oxide nanoparticles on the IONCNTs was observed by field emission transmission electron microscope (FE-TEM). Figure 2 presents an FE-TEM image including the electron diffraction (ED) patterns of IONCNT-10. The precipitated nanoparticles could be seen clearly on the surface of the IONCNTs and were amorphous as seen in Figure 2a. Their size ranged from approximately 10–20 nm. Numerous spots were observed in the ED patterns, as shown in the right upper area of Figure 2b, showing that the iron metal nanoparticles generated by the LPP reaction were polycrystalline.

X-ray photoelectron spectroscopy (XPS) is a commonly used analytical method for obtaining the chemical structure and state on the surface of the target materials and was used to confirm the chemical state of the IONCNTs. Figure 3 presents the narrow-range XP spectra of N and Fe of IONCNT-10. In the N 1s region as shown in Figure 3a, pyridinic and graphitic nitrogen were observed at 399.2 eV and 401.4 eV, respectively [31,32]. The amount of nitrogen determined by XPS (1.14 wt.%) was similar to the values obtained by EDS analysis (1.09 wt.%). This suggests that the nitrogen generated from the LPP reaction had combined successfully on the surface of the MWCNTs. Figure 3b shows the narrow XPS spectra of the Fe 2p region and peaks for Fe 2p_3/2_ and Fe 2p_1/2_ were observed at 711.9 eV and 725.4 eV, respectively. The spin-orbit separation between the two peaks was 13.5 eV and the iron precipitated on IONCNT was in the trivalent state. This indicates that the iron particles generated by the LPP reaction had precipitated in the form of Fe_2_O_3_ on IONCNT [33,34]. The atomic ratios of C 1s and O 1s of MWCNT measured by XPS analysis were 96.13% and 3.87%. The atomic ratios of C 1s, O 1s, N 1s and Fe 2p of IONCT-10 were quantified as 93.53%, 5.05%, 0.95%, and 0.47%, respectively. These results were similar to the quantitative analysis results of EDS analysis (in Table 1).

### 2.2. Electrochemical Measurements

Electrochemical performance of the as-fabricated CNTs by the LPP process was evaluated by the CV method and compared with those of the MWCNT. Figure 4 shows the current-voltage profiles of the MWCNTs and as-fabricated CNTs with different chemical compositions measured by CV at an applied voltage of 0.1–0.8 V and 10 mV/s of the potential scan rate. The CV curves of as-fabricated CNTs prepared using the MWCNT and LPP methods all exhibit a partially rectangular shape. On the other hand, the redox curves typical of metal oxide electrode materials did not appear in IONCNTs. The CV curve of the NCNTs was larger than that of the pristine MWCNTs due to nitrogen doping, which indicates improved electrochemical properties, such as energy storage capacity. In contrast, the IONCNTs exhibited superior electrochemical properties and the capacitance increased with increasing concentration of the iron precursor. The results show that the nitrogen dopant and iron oxide particles in the as-fabricated CNTs enhance the electronic conductivity and specific capacitance [35].

BET analysis was performed to investigate the specific surface area (SSA) of NCNT and IONCNTs prepared by LPP method. The SSA of MWCNT was measured to be 178.3 m^3^/g, and the SSA of NCNT and IONCNT-10 were measured to be 178.1 and 117.1 m^3^/g, respectively. There was almost no decrease in SSA by nitrogen doping, and about 0.6% of SSA was reduced in iron oxide-precipitated IONCNT-10. It is considered that SSA is changed due to the precipitation of iron oxide particle with a relatively small SSA compared to CNT.

The electrochemical stability of pristine MWCNTs and as-fabricated composites were evaluated by a galvanostatic charge-discharge process for 100 cycles. Figure 5 shows the cycling stability of the pristine MWCNTs and as-fabricated CNTs. The pristine MWCNTs exhibited a specific capacitance of 19.10 F/g at the first measurement and the remaining specific capacitance after 100 cycles were 16.50 F/g. The initial specific capacitance of the NCNTs, IONCNT-5, and IONCNT-10 prepared by the LPP reaction was 20.75, 22.15, and 22.76 F/g, respectively; all were higher than that of pristine MWCNTs, indicating less capacitance loss after repeated charge-discharge processes. After 100 charge-discharge cycles, the remaining specific capacitance of NCNTs, IONCNT-5, and IONCNT-10 were 18.90, 20.24, and 20.90 F/g. The loss yield of the as-fabricated CNTs ranged from 8.17% to 8.91%, which was significantly lower than that of the pristine MWCNTs (13.61%). It is attributed that the nitrogen and iron oxide introduced into the pristine MWCNTs increased the diffusion rate and charge transfer of ions.

Electrochemical impedance spectroscopy (EIS) was carried out to further characterize the behavior of the as-fabricated CNTs by the LPP reaction. Figure 6 presents the Nyquist plots of various half-coin cells made from the pristine MWCNTs and as-fabricated CNTs electrode. In the high frequency region, the semicircle represents the resistance between the electrolyte and the electrode of the coin cell, also referred to as the charge transfer resistance (Rct). An increase in the size of the semicircle means a decrease in capacitance quality. The Rct value of the NCNTs was 2.42 Ω and lower than that of the pristine MWCNTs (3.27 Ω). This was attributed to the high electronegativity of nitrogen atoms on the surface of the MWCNTs. The nitrogen atoms could generate sufficient defects and withdraw electrons to improve the kinetics of diffusion and transfer [36]. In addition, the Rct value of IONCNT-5 and IONCNT-10, which contained iron oxide nanoparticles, were 1.96 Ω and 1.84 Ω, respectively, because iron oxide particles promote charge transfer. The slope indicates capacitive behavior in the low frequency region, and the vertical-shaped slope represents pure capacitive behavior. The slope of the pristine MWCNTs was 11.3 and that of the as-fabricated CNTs was slightly higher. The slopes of the NCNT, IONCNT-5, and IONCNT-10 were 13.7, 15.4, and 16.1, respectively, indicating a higher diffusion rate of electrolyte ions inside the electrode materials.

## 3. Materials and Methods

### 3.1. Materials and Experimental Equipment

The multiwall carbon nanotubes (MWCNTs; K-Nanos-100P, Kumho Petrochemical Co. Ltd., Seoul, Republic of Korea) were used in this work. The diameter and density of the MWCNTs were 10–15 nm and 0.02–0.04 g/mL, respectively. Ammonium chloride (NH_4_Cl, Daejung Chemical & metals Co. Ltd., Siheung, Republic of Korea) and iron(II) chloride tetrahydrate (FeCl_2_·4H_2_O, Kanto chemical Co., Inc., Tokyo, Japan) were used as the precursor of nitrogen dopants and iron oxide particles. To induce dispersion, precipitated iron particles on the MWCNTs, cetrimonium bromide (CTAB, CH_3_(CH_2_)_15_N(CH_3_)_3_Br, Sigma-Aldrich, St. Louis, MO, USA) was used.

Figure 7 shows a schematic diagram of the LPP process introduced in this study to dope nitrogen element or precipitate iron oxide particles on MWCNTs. The LPP reaction system consisted of four parts: Power supply, cooling system, LPP reactor, and reactant tank. Two tungsten electrodes (ϕ 2 mm, 99.95% purity), insulated with a ceramic material, were placed in the middle of the LPP reactor made from Pyrex. The plasma field generated between the electrodes could be used to dope with nitrogen or precipitate iron oxide on the surface of MWCNT.

The reactant tank was a double tube type with cooling water circulating the outer channel. The cooling system was essential as the temperature of the reactant increased during the LPP reaction. The reactant solution was maintained at a constant temperature (293 K). The voltage, frequency, and pulse width applied to the electrodes from a power supply operating bipolar pulse type were 250 V, 30 kHz, and 5 μs, respectively. Detailed information of the LPP system is described elsewhere [37].

### 3.2. Preparation of NCNTs

NCNT composites were prepared by an LPP reaction and the details of preparation are as follows. First, 10 mol of NH_4_Cl, which is a precursor of nitrogen, was added to 600 mL of aqueous solution and stirred vigorously until dissolved completely. A 500 mg sample of MWCNTs was added to the reactant solution. The solution was ultrasonicated for 5 min and stirred for 1 h to produce a complete dispersion of MWCNTs. The reactant solution was transferred to a solution tank of the LPP process and the LPP reaction was then carried out for 1 h while circulating at a flow rate of 500 mL/min to dope on the surface of MWCNTs with nitrogen. After completion of the LPP reaction, the reaction products were centrifuged (10,000 rpm) and washed more than five times to separate the unreacted material and solvent. The separated products were dried in a vacuum oven for 24 h at 353 K.

### 3.3. Preparation of IONCNTs

IONCNTs were also prepared using a similar LPP process to that used for the fabrication of NCNTs. 500 mg of NCNTs were dispersed for 10 min in 600 mL of water, and 5 and 10 mM of iron chloride was put into the reactant solution to which NCNT had been dispersed and stirred until dissolved completely. CTAB, which was the dispersing agent of iron oxide particles, at a molar ratio of 50% with respect to the concentration of iron precursor was added to the reactant solution. The LPP reaction was generated in this reactant solution for 1 h to impregnate the NCNT surface with iron oxide particles. The reaction products from the LPP reaction were treated using the same method as for NCNT fabrication.

### 3.4. Characterization of Physicochemical and Electrochemical Properties

The physical and chemical properties of the as-fabricated composites were characterized using analytical instruments as FE-SEM (JSM-7100F, JEOL Ltd., Tokyo, Japan), FE-TEM (JEM-2100F, JEOL Ltd., Tokyo, Japan), and XPS (Multilab 2000 system, Thermo Fisher Scientific, Waltham, MA, USA). To assess the electrochemical performance of as-fabricated CNTs (NCNT and IONCNTs) fabricated by the LPP process, a coin cell was prepared as follows. A slurry was a homogeneous mixture consisting of 80 wt.% of the active materials (MWCNTs and as-fabricated CNTs), 10 wt.% of a conducting agent (Super-P carbon black (TIMCAL graphite & carbon com., Bironico, Switzerland)), and 10 wt.% of binder (polyvinylidene fluoride (PVDF)) in a N-methyl pyrrolidinone. The slurry was uniformly applied on the surface of Ni foil and dried in an oven for 24 h. The amount of the electrode materials loaded on to the Ni foil was 0.09–0.11 g. The 6 M KOH solution and glass felt with a 150 μm thickness were used as the electrolyte and the separator in this work. The prepared composite electrodes were placed on the top and bottom of the separator, and the electrolyte was filled and assembled using a mechanical coin-cell crimper. The electrochemical performance of MWCNT and as-fabricated CNTs was evaluated by cyclic voltammetry (CV) and electrochemical impedance spectroscopy (EIS). The actuation voltage, current density, and scan rate for CV was 0.1–0.8 V, 0.001 A/cm^2^, and 10 mV/s. The impedance spectroscopy measurement of the MWCNTs and as-fabricated CNTs were performed within the frequency range, 0.01 to 10 kHz. All electrochemical properties were measured using a VSP Potentiostat (Bio-logic Science Instruments, Grenoble, France).

## 4. Conclusions

NCNTs and IONCNTs as potential electrode materials of supercapacitors were prepared by the LPP process. FE-SEM confirmed that nitrogen and iron were dispersed evenly over the surface of the pristine MWCNTs and the amount of iron oxide particles increased with increasing the initial precursor concentration. XPS showed that in the NCNTs, nitrogen was combined with MWCNTs in form of pyridinic and graphitic nitrogen. In addition, iron oxide particles on the surface of IONCNTs existed in the form of Fe_2_O_3_. The average size of the amorphous iron oxide particles precipitated evenly over the MWCNT surface ranged from 10 nm to 20 nm. The obtained NCNTs and IONCNTs using the LPP process not only showed a higher initial specific capacitance but also maintained superior cycling stability compared to pristine MWCNT. The specific capacitance of the IONCNTs increased with increasing iron precursor concentration.

## Figures and Tables

**Figure 1 ijms-19-03830-f001:**
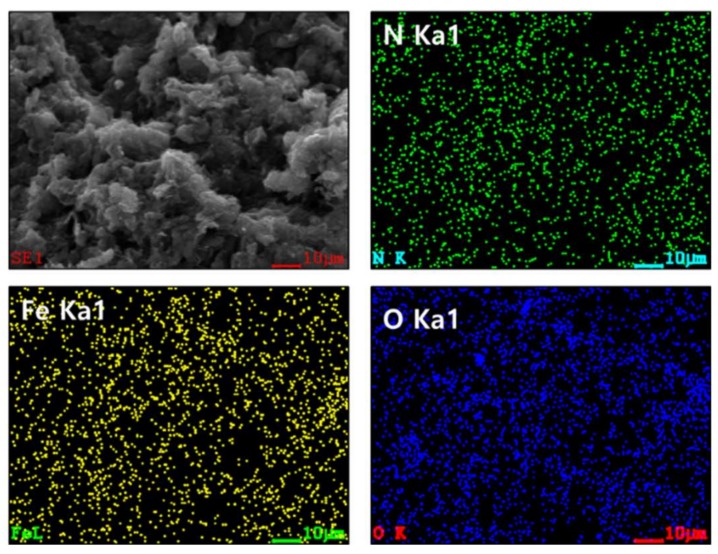
Field emission scanning electron microscope (FE-SEM) real photograph and elemental mapped results with nitrogen, iron, and oxygen element of iron oxide particles precipitated on nitrogen-doped carbon nanotubes (IONCNT-10) prepared by liquid phase plasma (LPP) process.

**Figure 2 ijms-19-03830-f002:**
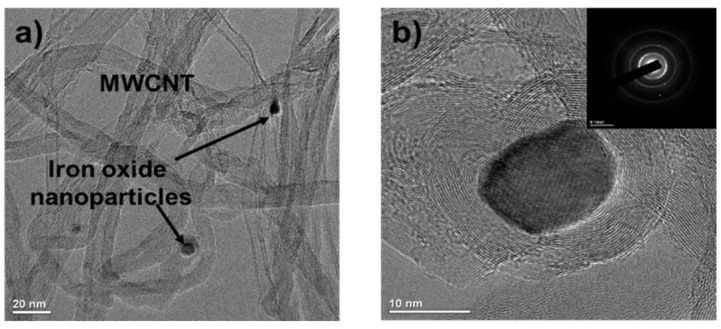
Field emission transmission electron microscope (FE-TEM) (**a**) and HR-FETEM image (**b**) including electron diffraction (ED) patterns of IONCNT-10.

**Figure 3 ijms-19-03830-f003:**
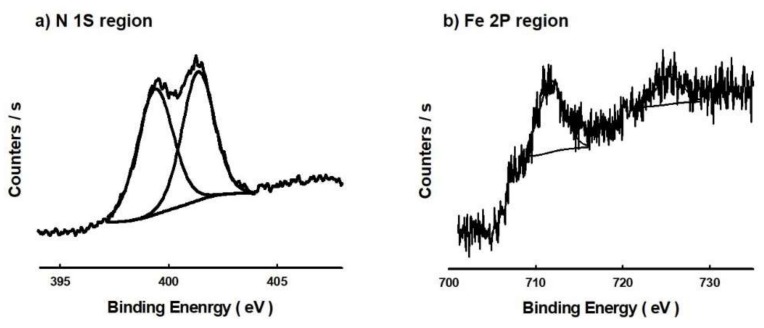
High-resolution X-ray photoelectron spectroscopy (XPS) spectra for the N 1S region (**a**) and Fe 2p region (**b**) of IONCNT-10 prepared by LPP process.

**Figure 4 ijms-19-03830-f004:**
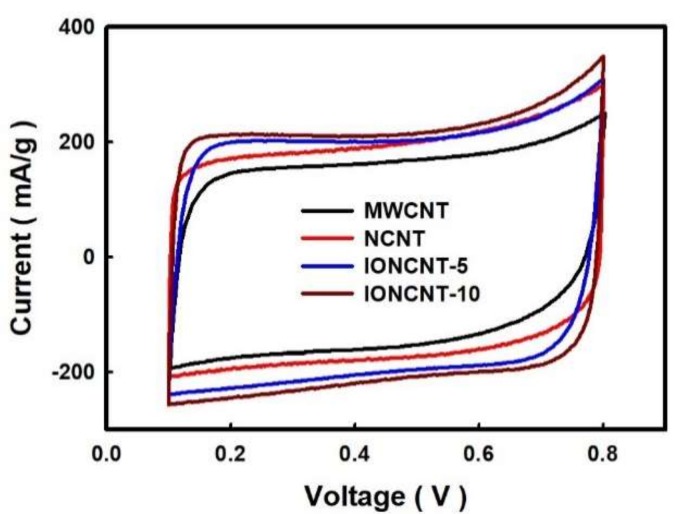
Cyclic voltammetry (CV) curves of MWCNT and as-fabricated CNTs by LPP process.

**Figure 5 ijms-19-03830-f005:**
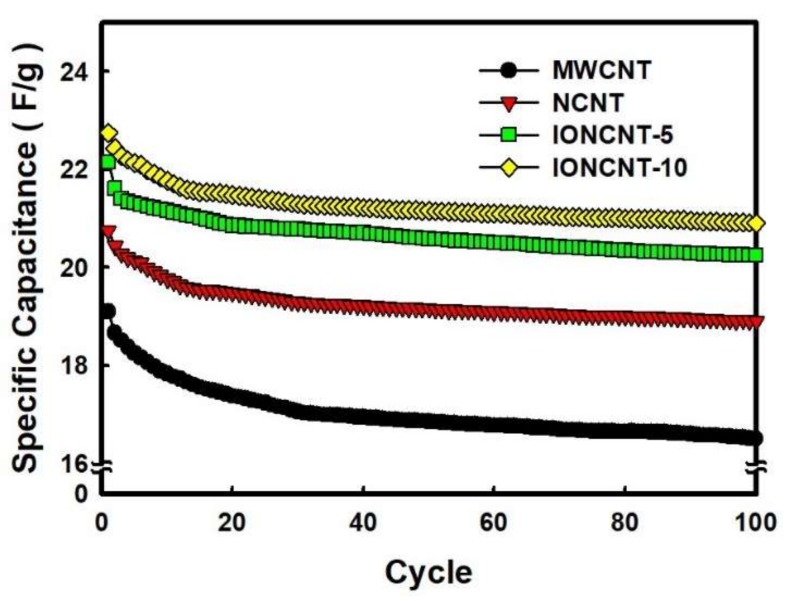
The specific capacitance of pristine MWCNTs and as-fabricated by LPP process with different precursor concentrations.

**Figure 6 ijms-19-03830-f006:**
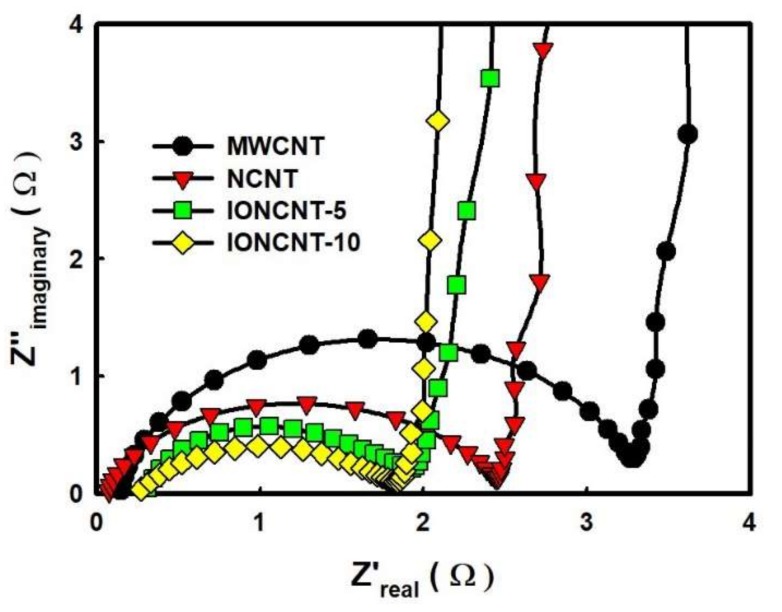
Electrochemical impedance spectroscopy (Nyquist plots) of pristine MWCNTs and as-fabricated CNTs.

**Figure 7 ijms-19-03830-f007:**
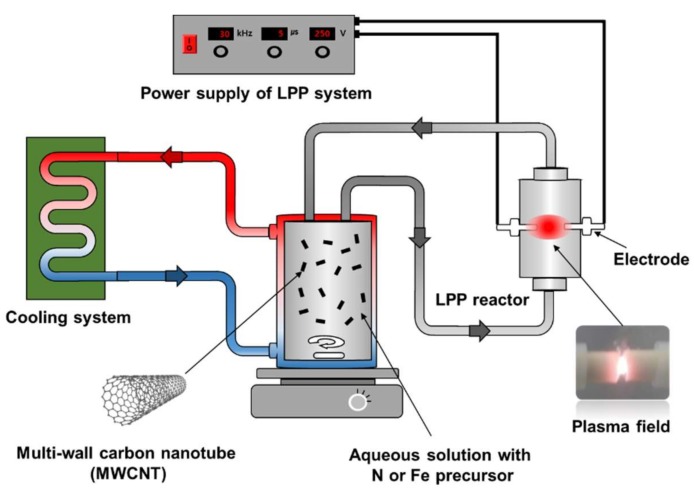
Schematic diagram of the LPP system for preparation of NCNT and IONCNTs.

**Table 1 ijms-19-03830-t001:** Chemical composition of the multiwall carbon nanotube (MWCNT) and as-fabricated carbon nanotubes (CNTs) by the LPP process.

Samples	Carbon		Oxygen		Nitrogen		Iron	
	Wt.%	At.%	Wt.%	At.%	Wt.%	At.%	Wt.%	At.%
MWCNT	94.94	96.15	5.06	3.85	0.00	0.00	0.00	0.00
NCNT	93.09	94.57	5.42	4.14	1.49	1.30	0.00	0.00
IONCNT-5	91.61	94.02	6.03	4.65	1.23	1.08	1.13	0.25
IONCNT-10	90.48	93.57	6.47	5.03	1.09	0.97	1.96	0.44

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
