# Peer review of "Enhanced Electrochemical Performance of Carbon Nanotube with Nitrogen and Iron Using Liquid Phase Plasma Process for Supercapacitor Applications"

_ijms, 2018, doi:10.3390/ijms19123830_

Round 1

Reviewer 1 Report

The presented topic is interesting and very well presented. The high quality of the content makes the paper as an important step in the field of supercapacitors research. The research methods clearly described. Good quality of the paper but a few suggestions concerning the style:

1. line 31 “long-term” is suggested

2. line 69 did you mean IONCNT-10 and IONCNT-5. Did you mean process using 5 mM of iron chloride? A few words of explanation is needed between lines 69 and 94

4. line 135 I think “The initial specific capacitance…. WAS 20.75”

5. line 176 I think should be “because OF the temperature”

6. Enlarge Figure 7 a little to unify font size used in figures

Author Response

Thank you for allowing us the opportunity to respond to reviewer comments regarding the above-referenced manuscript. The manuscript was revised according to the reviewer’s comments. Changes made in response to the comments are described below.

1. line 31 “long-term” is suggested.

[Response] As recommended, we have revised sentence.

2. line 69 did you mean IONCNT-10 and IONCNT-5. Did you mean process using 5 mM of iron chloride? A few words of explanation is needed between lines 69 and 94.

[Response] A description of IONCNT-10 and IONCNT-5 has been added to the text. (line 80-81)

3. line 135 I think “The initial specific capacitance…. WAS 20.75”

[Response] As recommended, we have revised. (line 137)

4. line 176 I think should be “because OF the temperature”.

[Response] As recommended, we have revised. (line 179)

5. Enlarge Figure 7 a little to unify font size used in figures.

[Response] As recommended, the font in the figure was modified, and the size was enlarged.

Reviewer 2 Report

The manuscript “Enhanced electrochemical performance of carbon nanotube with nitrogen and iron using liquid phase plasma process for supercapacitor applications” synthesized the iron oxide particles precipitated on nitrogen-doped carbon nanotubes (IONCNTs) by a liquid phase plasma (LPP) method. The authors claimed that the nitrogen element and amorphous iron oxide nanoparticles were evenly disseminated on the pristine multiwall carbon nanotubes (MWCNTs). The electrochemical performance of the composites was improved with introduced nitrogen and iron oxide. The authors have provided solid data to back up the conclusions in most cases. However, when reading the manuscript some questions arise, therefore some complementary information and revision should be taken into account before being published in “International Journal of Molecular Sciences”.

1. In line 70, the iron element was indicated by yellow dots, not red.

2. Is there any chlorine element can be found from the EDS of IOCNT-10? Since the abundance of chlorine element in the reagents.

3. In line 79, the sentence “The number located at the back of the sample code means the concentration of iron precursor in aqueous solution.” is unclear and confusing. Facile designations of IONCNT-5 and IONCNT-10 would be clearer.

4. In Table 1, EDS is unqualified for precisely quantitative elemental analysis. The elemental amounts by XPS are recommended to analyze the chemical compositions.

5. In the CV section, the authors claimed that “In all curves, the shape of the curves exhibited a partially rectangular shape without a redox curve, which is the typical characterization of metal oxide electrode materials.” However, there is not metal oxide in MWCNT and NCNT.

6. In Figure 4, the reference electrode should be given in the title of X-axis.

7. In Figure 6, the title of vertical axis is incorrect. Moreover, the coordinate range of X and Y axis should be consistent.

8. In line 163, the expression “Carbon-based materials were used as the multiwall carbon nanotubes” is unclear and confusing.

9. In section 3.4, the details about the half-coin cell are required, such as the mass loading of the electrode, the counter and reference electrodes, the assembling procedure, and so forth.

10. The BET characterization is suggested for the comparison of surface area.

The English writing in manuscript needs to be checked carefully since the meaning of some sentences cannot be understood by the improper use of English or ambiguous description.

Author Response

Thank you for allowing us the opportunity to respond to reviewer comments regarding the above-referenced manuscript. The manuscript was revised according to the reviewer’s comments. Changes made in response to the comments are described below.

1. In line 70, the iron element was indicated by yellow dots, not red.

[Response] As recommended, we have revised yellow. (line 71)

2. Is there any chlorine element can be found from the EDS of IOCNT-10? Since the abundance of chlorine element in the reagents.

[Response] Iron chloride, the precursor of iron oxide particles, is ionized into Fe2+ and Cl- in the reactant solution, and Fe2+ is reduced by LPP reaction and deposited on the surface of NCNT as iron particles. On the other hand, Cl- is present as a non-reactant in the reactant solution and composites. In this study, composites synthesized by LPP method were washed several times to remove impurities such as Cl, which were confirmed by EDS analysis.

3. In line 79, the sentence “The number located at the back of the sample code means the concentration of iron precursor in aqueous solution.” is unclear and confusing. Facile designations of IONCNT-5 and IONCNT-10 would be clearer.

[Response] A description of IONCNT-10 and IONCNT-5 has been added to the text. (line 80-81)

4. In Table 1, EDS is unqualified for precisely quantitative elemental analysis. The elemental amounts by XPS are recommended to analyze the chemical compositions.

[Response] As you pointed out, quantitative elemental analysis using EDS is insufficient. But we used our analysis equipment as much as possible. Reliability by repeated analysis, and XRD analysis together to cross-check the data.

5. In the CV section, the authors claimed that “In all curves, the shape of the curves exhibited a partially rectangular shape without a redox curve, which is the typical characterization of metal oxide electrode materials.” However, there is not metal oxide in MWCNT and NCNT.

[Response] As recommended, the corresponding sentence was modified as follows:

“The C-V curves of as-fabricated CNTs prepared using the MWCNT and LPP methods all exhibit a partially rectangular shape. On the other hand, the redox curves typical of metal oxide electrode materials did not appear in IONCNTs” (line 122-125)

6. In Figure 4, the reference electrode should be given in the title of X-axis.

[Response] Electrochemical properties were measured by preparing full-coin cells. Since CNT was used as a cathode and an anode electrode, a separate reference electrode was not needed. Coin cell manufacturing method was revised and supplemented. (line 212, 216-220)

7. In Figure 6, the title of vertical axis is incorrect. Moreover, the coordinate range of X and Y axis should be consistent.

[Response] As recommended, the title and scale of the vertical axis in Figure 6 have been revised.

8. In line 163, the expression “Carbon-based materials were used as the multiwall carbon nanotubes” is unclear and confusing.

[Response] The line 163 sentence was revised as follows:

“The multiwall carbon nanotubes (MWCNTs; K-Nanos-100P, Kumho Petrochemical co. ltd., Seoul, Republic of Korea) were used in this work.” (line166-167)

9. In section 3.4, the details about the half-coin cell are required, such as the mass loading of the electrode, the counter and reference electrodes, the assembling procedure, and so forth.

[Response] The coin cells prepared in this study do not use counter and reference electrodes unlike the three electrode system. Coin cell manufacturing method was revised and supplemented. (line 216-220)

10. The BET characterization is suggested for the comparison of surface area.

[Response] The specific surface areas of CNT and IONCNTs used in this study were measured. But I could not find any change, so I did not give any results. This is considered to be because the amount of added metal is very small.

11. The English writing in manuscript needs to be checked carefully since the meaning of some sentences cannot be understood by the improper use of English or ambiguous description.

[Response] The manuscript was revised and proofread by an English editor according to the reviewer’s comment. The editing certificate is attached below:

Round 2:

Reviewer 2 Report:

Almost all the questions and comments raised by the reviewers were legitimately explained and revised. The accuracy and detail of the manuscript were also improved further after revision. However, a few questions are still there and needed further revision.

1. The reviewer insists that the quantitative elemental analysis using XPS would be better. It should not be difficult since the XPS measurement has already been carried out for the samples.

2. The authors mentioned the quantitative elemental analysis was cross-checked by XRD analysis. However, there are not XRD results shown in the manuscript. Actually, it is also worthy to do XRD tests for the samples to characterize the structure of the iron oxide particles.

3. The authors claimed that they could not find any change from the specific surface areas of CNT and IONCNTs, so they did not give any results. However, this is exactly the reason why the reviewer asked for the BET characterization. It should be proven that the higher capacity of IONCNTs comes from the iron oxide particles rather than higher surface areas. The BET results and related discussion should be provided.

Author Response

The manuscript was revised according to the reviewer’s comments. Changes made in response to the comments are described below.

1. The reviewer insists that the quantitative elemental analysis using XPS would be better. It should not be difficult since the XPS measurement has already been carried out for the samples.

[Response] As recommended, the results of XPS quantitative analysis of IONCNT-10 shown in Fig. 3 were added to the text. (line 115~118)

2. The authors mentioned the quantitative elemental analysis was cross-checked by XRD analysis. However, there are not XRD results shown in the manuscript. Actually, it is also worthy to do XRD tests for the samples to characterize the structure of the iron oxide particles.

[Response] XPS analysis was misspelled by XRD analysis in the previous answer. I am sorry about the confusion in judging. We actually performed XRD analysis of IONCTs. However, no change was observed in the XRD analysis since the amount of iron contained in the IONCTs was very small, less than 1 At. %. For this reason, XRD results are not presented in this manuscript.

3. The authors claimed that they could not find any change from the specific surface areas of CNT and IONCNTs, so they did not give any results. However, this is exactly the reason why the reviewer asked for the BET characterization. It should be proven that the higher capacity of IONCNTs comes from the iron oxide particles rather than higher surface areas. The BET results and related discussion should be provided.

[Response] As recommended, BET analysis results are added to the text. (line 133-138)

Thank you again for the opportunity to respond to reviewer comments. Please contact me if you have any questions. I can be reached anytime via email at [email protected] or during the day through office at +82-61-750-3814.

Sincerely,

Sang-Chul Jung, Ph.D.

Professor

Department of Environmental Engineering

Sunchon National University